# Chemical Fingerprinting of Cryptic Species and Genetic Lineages of *Aneura pinguis* (L.) Dumort. (Marchantiophyta, Metzgeriidae)

**DOI:** 10.3390/molecules26041180

**Published:** 2021-02-22

**Authors:** Rafał Wawrzyniak, Wiesław Wasiak, Beata Jasiewicz, Alina Bączkiewicz, Katarzyna Buczkowska

**Affiliations:** 1Faculty of Chemistry, Adam Mickiewicz University in Poznań, Uniwersytetu Poznańskiego 8, 61-614 Poznań, Poland; wasiakw@amu.edu.pl (W.W.); beatakoz@amu.edu.pl (B.J.); 2Faculty of Biology, Adam Mickiewicz University in Poznań, Uniwersytetu Poznańskiego 6, 61-614 Poznań, Poland; alinbacz@amu.edu.pl (A.B.); androsac@amu.edu.pl (K.B.)

**Keywords:** *Aneura pinguis*, liverworts, cryptic species, genetic lineages, chemical markers, HS-SPME/GC-MS, DNA barcode, volatile compounds, multivariate analysis

## Abstract

*Aneura pinguis* (L.) Dumort. is a representative of the simple thalloid liverworts, one of the three main types of liverwort gametophytes. According to classical taxonomy, *A. pinguis* represents one morphologically variable species; however, genetic data reveal that this species is a complex consisting of 10 cryptic species (named by letters from A to J), of which four are further subdivided into two or three evolutionary lineages. The objective of this work was to develop an efficient method for the characterisation of plant material using marker compounds. The volatile chemical constituents of cryptic species within the liverwort *A. pinguis* were analysed by GC-MS. The compounds were isolated from plant material using the HS-SPME technique. Of the 66 compounds examined, 40 were identified. Of these 40 compounds, nine were selected for use as marker compounds of individual cryptic species of *A. pinguis*. A guide was then developed that clarified how these markers could be used for the rapid identification of the genetic lineages of *A. pinguis*. Multivariate statistical analyses (principal component and cluster analysis) revealed that the chemical compounds in *A. pinguis* made it possible to distinguish individual cryptic species (including genetic lineages), with the exception of cryptic species G and H. The classification of samples based on the volatile compounds by cluster analysis reflected phylogenetic relationships between cryptic species and genetic lineages of *A. pinguis* revealed based on molecular data.

## 1. Introduction

The genus *Aneura* Dumort. (Metzgeriidae) is a representative of the liverworts, a phylum of small terrestrial plants that has played an important role in the evolution of early land plants. Liverworts were among the first plants to colonize land [1,2], and they have been documented in the fossil record as early as 470 million years ago [3,4]. Today, liverworts are an important component of many terrestrial ecosystems [5]. Liverworts are also the source of a large number of biologically active compounds, such as terpenoids and aromatic compounds, synthesized and accumulated in oil bodies [6,7,8]. Many of these compounds occur only in liverworts and exhibit interesting biological activities [9,10,11,12,13,14]. These compounds are also a valuable source of markers for resolving taxonomic problems and identifying individual species [15,16,17,18,19,20]. Revealing differences in the composition of chemical compounds between closely related species is important not only for the identification of species but also for the clarification of evolutionary relationships between them [19,21,22,23].

*Aneura pinguis* (L.) Dumort., a common and widely distributed species, is an example of a simple thalloid liverwort, one of the three main types of liverwort gametophytes [24,25,26,27,28]. *Aneura pinguis*, although taxonomically treated as one species, from the biological point of view it is a complex of cryptic species, i.e., species which, in the absence of morphological differences, are reproductively isolated and have separate gene pools [29,30,31,32]. Recent molecular studies have revealed the existence of 10 cryptic species of *A. pinguis* that have been tentatively named with letters from A to J, genetic divergence between them in DNA sequences ranged from 1.45% to 7.41% [33]. Four of these species (A, B, C and E) are further subdivided into evolutionary lineages: species A and B into three lineages each (A1, A2, A3 and B1, B2, B3, respectively) and species C and E into two lineages each (C1, C2 and E1, E2, respectively)[33]. In total, 16 evolutionary lineages have been identified within *A. pinguis* at the molecular level. Eight cryptic species (14 evolutionary lineages) occur in Poland, and two species are not found in this region: species D discovered in Great Britain and species J limited exclusively to Japan [33]. Unfortunately, this high genetic diversity is not reflected in the presence of diagnostic features therefore individual cryptic species cannot be distinguished in the traditional way [31,32,33]. A helpful tool in identifying these species may be the analysis of the composition of chemical compounds.

Recently, chemotaxonomic studies of the genetically distinct cryptic species of *A. pinguis* revealed the presence of compounds specific to four cryptic species of *A. pinguis* (A, B, C and E) and *A. maxima* (Schiffn.) Steph. [34]. Differences in the composition and amount of volatile compounds were also found for evolutionary lineages distinguished based on molecular studies of species A (A1, A2, A3) and species B (B1, B2, B3). Thus, there is a close correlation between the chemical composition of the plant and its genotype [35]. 

The aim of this study was to determine the contents of volatile compounds in species C (lineages C1, C2), E (lineages E1, E2), G, H and I that have been recognized by molecular studies but whose chemical composition has not yet been analysed. In addition, we aimed to compare the chemical composition of genetically recognized cryptic species. Our ultimate goal was to evaluate the utility of the detected compounds as chemotaxonomic markers for identifying the cryptic species and evolutionary lineages of *A. pinguis*.

## 2. Results and Discussion

### 2.1. Chemical Profile

Thirty seven samples of *A. pinguis* belonging to seven different genetic lineages were subjected to chemical analyses. Volatile compounds were determined for the following species and genetic lineages (C1, C2, E1, E2, G, H and I) recognized based on molecular studies [33]. Details on locality and collection time of the samples are shown in Table 1. In the multivariate statistical analysis, results obtained in the present studies were compared with all remaining genetic lineages from previous research on *A. pinguis* [34,35]. As in previous studies, a small amount of biological material that could be collected for individual genetic lineages of *A. pinguis* from a given habitat in different vegetation seasons was determined using the HS-SPME technique.

Results of the chemical analyses are shown in Table 2, Table 3, Table 4 and Table 5. The tables also contain compounds of no. 9, 11, 57 and 58 provided in a previous study [35]. These data facilitate comparisons of the current results with those previously described. 

To determine whether the chemical compounds can act as chemotaxonomic markers permitting the determination of all cryptic species of *A. pinguis*, the data obtained for samples in current studies (C1, C2, E1, E2, G, H, I) were compared in multivariate statistical analyses with data on the remaining genetic lineages of *A. pinguis* previously published [35]. Two classification methods were used: principal component analysis (PCA) and cluster analysis (CA). Both analyses were performed using all 66 volatile compounds detected in the examined *A. pinguis* samples. 

In the cluster analysis, the classification of samples based on the volatile compounds largely reflected the phylogenetic relationships between cryptic species and genetic lineages of *A. pinguis* proposed by Bączkiewicz et al. [33] using molecular data. The chemical compounds found in *A. pinguis* make it possible to distinguish the vast majority of cryptic species and genetic lineages of the species (Figure 1). On the dendrogram constructed based on the Euclidean distance and Ward’s linkage method, 11 clusters can be recognized based on the Mojena index, which basically matched with the genetic lineages of *A. pinguis*, with the exception of two cryptic species: G and H, which form one cluster. The analysed samples of *A. pinguis* were grouped into two large clusters: the first cluster included species B, C, F, G, H, I and lineage E2, and the second cluster included A and lineage E1. All genetic units analysed in the present study (C1, C2, E1, E2, G, H and I) differed from the other species of *A. pinguis* (A, B and F) in the composition of chemical compounds, accounting for the fact that species A and B are divided into the genetic lineages A1, A2, A3 and B1, B2, B3, respectively. Samples of species C form two closely related clusters corresponding to the genetic lineages C1 and C2 based on molecular studies [33], however, these clusters differ to a lesser extent, like B2, B3, thus, on the basis of the Mojena’s index, they do not form separate clusters. In contrast, the two genetically closely related lines of species E (E1 and E2) show a high degree of distinctiveness in their chemical compounds [33]. Samples of lineage E1 formed a distinct group that was placed in one cluster with species A, which was similar to results of genetic studies; however, samples of E2 were in a second cluster that included species B, C, F, G, H, and I. The detected chemical compounds poorly differentiate between species G and H, whose distinction is supported by molecular studies [33].

To verify the patterns in diversity observed in the cluster analysis and express these data more simply, a PCA was conducted on the volatile compound data. PCA on the 66 compounds produced eight significant principal components (PCs) that explained 98.2% of the variation (R2X) and 75.4% the predicted variation (Q2). Figure 2 shows the three-dimensional scatter plot of the first three principal components, which explained 91.9% of the total variance included in the analysed volatile compounds. The PCA of the examined samples of *A. pinguis* revealed the existence of 13 groups of samples corresponding to the genetic lineages identified by DNA markers, because species G and H could not be distinguished. The PC1 axis primarily divided species B (genotypes B1, B2, B3) and F from G, H, I, A3 and E1 based on the relative content of compound 52, which made the largest contribution in PC1. The PC2 axis separated lineage A1 and A2 from A3 and lineage B3 from B1, B2 and F, while the PC3 axis differentiated lineage A1 from A2 and lineages B1, B3 and F from I and E1, which was the most distinct lineage in the PCA score plot. In contrast, the E2 genotype did not show strong differentiation from A1 and A2 (Figure 2). In the PC2 axis, compound 40 made the largest contribution, while in the PC3 axis, compounds 60 and 61 made the largest contributions; thus, these compounds played the most important role in separating genotypes across PC2 and PC3. From both statistical analyses (CA and PCA) is evident, that most species and genetic lineages of *A. pinguis* can be distinguished on the basis of their chemical composition. The differentiation and the degree of affinity between them revealed on the basis of chemical studies is almost completely consistent with the results of molecular analyzes [33]. An interesting exception is species E, lineages E1 and E2, which in terms of DNA sequences are closely related and belong to one clade (genetic divergence is equal to 0.38%), showing great diversity in the composition of chemical compounds and form strongly distinct sets. On the other hand, for the pair of species G and H, we observed small differences in the composition of chemical compounds, despite a significant molecular difference (2.47%) between them [33].

The 66 volatile compounds were determined based on chemical tests carried out on liverworts within *A. pinguis*, 40 of which were identified. The data collected in this paper and the work published by Wawrzyniak et al. [35] indicated that the composition of volatile compounds was primarily determined by genetics. Season and habitat appeared to have no effect on the chemical composition of compounds produced. The identified metabolites were compounds belonging to groups of oxygen derivatives of sesquiterpenes, sesquiterpenes, monoterpenes, aliphatic hydrocarbons and aromatic hydrocarbons. The observed differences in the composition of metabolites for individual *A. pinguis* genetic lineages were so salient that they could be used to identify biological material. Quite similar, Ludwiczuk et al. [21], demonstrated the use of volatile compounds as chemical markers for the identification of *Conocephalum conicum* cryptic species. The importance of the composition of volatile compounds as chemical markers is confirmed by the results of numerous studies of different liverwort species [8,18,19,20,21,22,23].

To simplify identification via metabolite composition, nine compounds from the group of 66 compounds examined that best differentiated the analysed material were selected that permit plant material to be assigned to specific species and even genetic lineages. Data on the content of these compounds for individual genetic lineages and species in the radar charts are shown in Figure 3.

### 2.2. Guide for the Identification of Cryptic Species and Genetic Lineages

These results also allowed us to prepare a guide describing a procedure through which different genetic lineages could be identified based on the content of marker compounds. The first step for identifying material is to characterize the content of pinguisone. The content of this compound can be used to classify the tested material into four groups (below 20%, 20–50%, 50–70% and above 70%). The next step is to determine the content of dihydrobryopterin A, which identifies lineages A3 and E1. Other groups can be identified based on the content of deoxopinguisone (species I) and costunolide (lineages C1, C2, B1, and species G, H). The absence of the compound characterized by IR = 1934 permits the identification of E2. The content of the compound characterized by IR = 1989 permits the differentiation of A1 and A2. The identification of lineages B2, B3, C2, and species F is based on the content of compounds characterized by IR = 1533 and IR = 1633. As seen in Figure 4, species G and H cannot be distinguished based on their chemical composition. The quantitative differences in the detected compounds are obviously not enough for identification of species G or H (e.g., compound IR = 1533 in Table 5).

## 3. Materials and Methods

### 3.1. Plant Material

Plant material included 37 samples of *Aneura pinguis* (L.) Dumort. (Aneuraceae) collected in different regions of Poland (Table 1). Some of the analysed genotypes of *A. pinguis* have small thallus sizes and often grow in small colonies composed of only a few thalli [32]. Several selected samples were collected repeatedly from the same place in consecutive years and at different times of the year (Table 1). From each sample, a portion was dried and deposited as a voucher in the POZW Herbarium. The remaining portion of the sample was cleaned and used for analyses. Samples for DNA and chemical analyses were stored at −30 °C. Chemical analyses were performed on samples of *A. pinguis* identified to species and genetic lineage based on six DNA barcodes; sequences were deposited in GenBank [33]. These sequences can be obtained from GenBank using the POZW numbers provided in Table 1.

### 3.2. GC Analysis of Volatile Compounds

The analysis of the volatile compounds was performed using a previously described GC-MS method [34,35]. Conditions of sorption and desorption were optimized through selection of the type of stationary phases, coated fibres, the amount of biological material, time, and temperature. Frozen 10 mg samples of *A. pinguis* were placed in a screw-capped vial with a 1.7 mL silicone/Teflon membrane. Next, the vial was heated at 50 °C, and headspace solid-phase micro-extraction was carried out for 60 min. SPME-fused silica fibres coated with divinylbenzene/carboxen/polydimethylsiloxane (DVB/CAR/PDMS) stationary phases were used (Sigma-Aldrich, St. Louis, MO, USA). Desorption was performed at 250 °C for 10 min. The above operations were carried out by the autosampler TriPlus RSH (Thermo Fisher Scientific, Waltham, MA, USA). Compounds isolated from the biological material were analysed by a gas chromatograph Trace 1310 coupled with a mass spectrometer ISQ QD (Thermo Fisher Scientific). The mass selective detector was operated at 70 eV in the EI mode over the *m*/*z* range 30–550 and temperature transfer line 250 °C. The GC instrument was equipped with a capillary column coating with the silphenylene phase (Quadex 007-5MS: 30 m × 0.32 mm i.d., film thickness 0.25 μm; Bethany, CT, USA). Helium was used as the carrier gas at a flow rate of 1.0 mL/min. The oven temperature was programmed to change from 60 to 230 °C at 4 °C/min and then remain stable at 230 °C for 40 min. Each sample was analysed three times.

After compound determination, the desorption procedure and GC analysis were repeated one more time. The lack of any peaks indicated that the selected conditions allowed the efficiency of the desorption process to reach 100%. The constituents were identified by comparing their MS spectra with those from the literature, reference compounds, computer matching against the NIST 11, data obtained from NIST Chemistry WebBook databases, Mass Finder 4 library, Adams library databases and Pherobase databases [36,37,38,39]. The identification of compounds was verified by Kovats retention indices. Kovats retention indices were determined relative to a homologous series of *n*-alkanes (C8–C26) under the same operating conditions. The relative concentrations of the components were obtained by peak area normalization without applying correction factors. The GC-MS analysis was performed in a scan mode; compounds were recorded via a TIC (total ion chromatogram). Table 2, Table 3, Table 4 and Table 5 show the data for compounds with concentrations greater than 0.01%. Recorded compounds below the 0.01% threshold were not included in the tables.

### 3.3. Statistical Analysis

Multivariate statistical data obtained in the current study for samples of lineages C1, C2, E1, E2, and species G, H, I were compared with data from the remaining genetic lineages of *A. pinguis* to determine whether the composition of volatile compounds differed within the group of cryptic species of *A. pinguis* that were studied [35]. Two classification methods were used: cluster analysis and principal component analysis. In cluster analysis, hierarchical clustering based on Euclidean distances according to Ward’s linkage method was used [40]. Principal component analysis was based on the covariance matrix of all 66 compounds using the nonlinear iterative partial least squares algorithm for constructing the PCA model. The v-fold method was used to find the optimal number of principal components that reached the maximum Q2. Statistical significance of the principal components was assessed based on the rule, Q2 > Limit. Statistical analyses were performed using STATISTICA 13.3 (StatSoft, Kraków, Poland).

## 4. Conclusions

GC-MS analysis of the volatiles revealed variability in the chemical composition of the various cryptic species of *A. pinguis*. The composition of volatile compounds in this species is largely genetically determined. The current and previous studies suggest that the selection of marker compounds might permit the rapid identification of almost all cryptic species of *A. pinguis*. These compounds include pinguisone (**52**) and its derivatives: deoxopinguisone (**40**), dihydrobryopterin A (**55**), and methyl norpinguisonate (**56**). Based on the content of these compounds, lineages E1 can be distinguished from A3 and I. The content of these compounds also permits the remaining lineages to be distinguished. The compounds IR = 1533 (**37**) and IR = 1633 (**46**) permit the differentiation of lineages B2, B3, C2 and species F. Costunolide (**60**) appears to be a robust chemical marker for lineage B1, C1, C2, and species G, H. The compound IR = 1934 (**61**) identifies lineage E2. Compound IR = 1989 (**65**) identifies lineage A1 and A2. The role of chemical compounds as chemotaxonomic markers for cryptic species of *A. pinguis* was supported by two multivariate statistical methods (CA and PCA), which grouped the studied samples based on the detected chemical compounds and was largely consistent with the genetic determination based on DNA barcoding sequences.

## Figures and Tables

**Figure 1 molecules-26-01180-f001:**
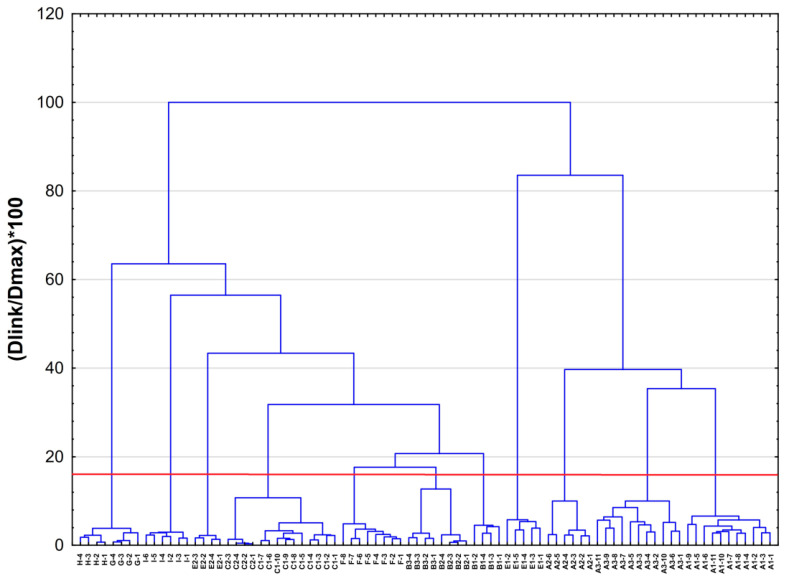
Dendrogram of samples of cryptic species of *A. pinguis* (C, E, G, H, and I) compared with the remaining species of *A. pinguis* examined in a previous study [35]. The dendrogram was constructed based on the Euclidean distance according to Ward’s linkage method. The dendrogram was based on all of the detected compounds in the samples. The transverse line indicates the intersection of the dendrogram according to the Mojena index.

**Figure 2 molecules-26-01180-f002:**
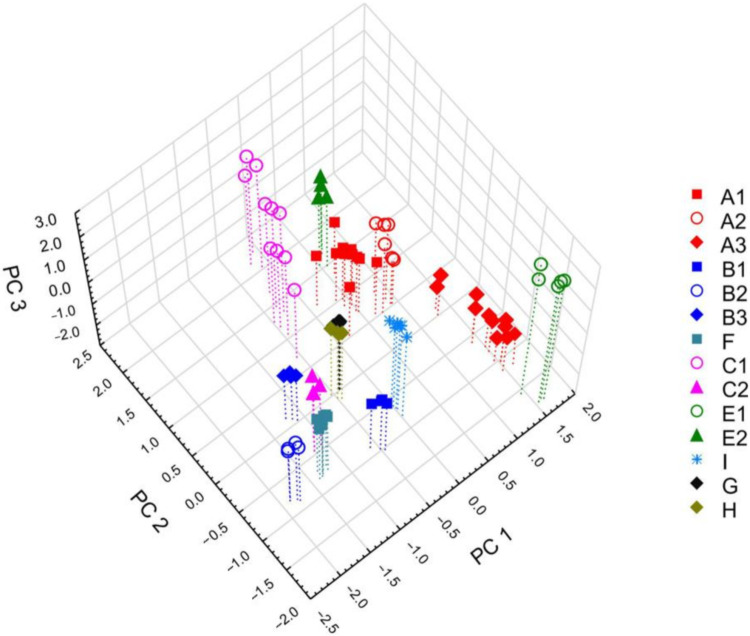
Three-dimensional PCA score scatter plot based on all of the detected compounds in samples of cryptic species of *A. pinguis* (C, E, G, H, and I). The percentage of explained variance (R^2^X) was 81.4% for PC1, 6.2% for PC2, and 4.3% for PC3, and the predictive ability (Q^2^) was 36.7%, 16.7%, and 19.9%, respectively. Species in this study were compared with the remaining species of *A. pinguis* examined in a previous study [35].

**Figure 3 molecules-26-01180-f003:**
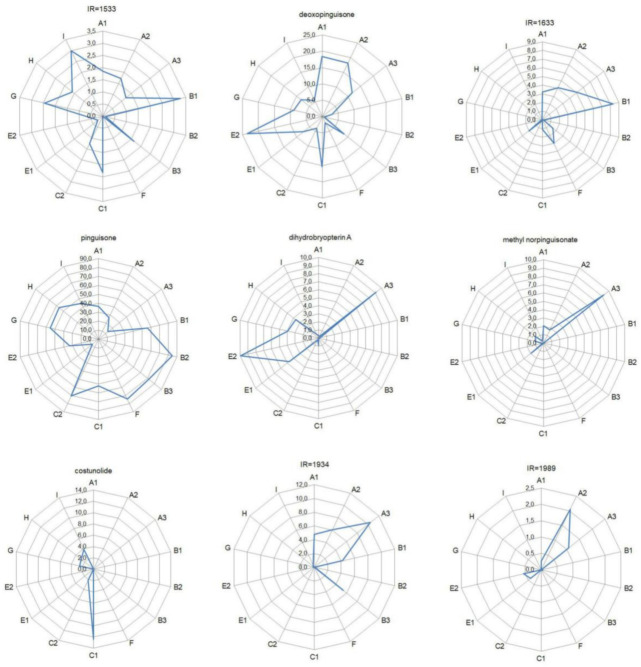
Radar chart for marker compounds used to identify genetic lineages. The contents of compounds for individual genetic lineages are given in %.

**Figure 4 molecules-26-01180-f004:**
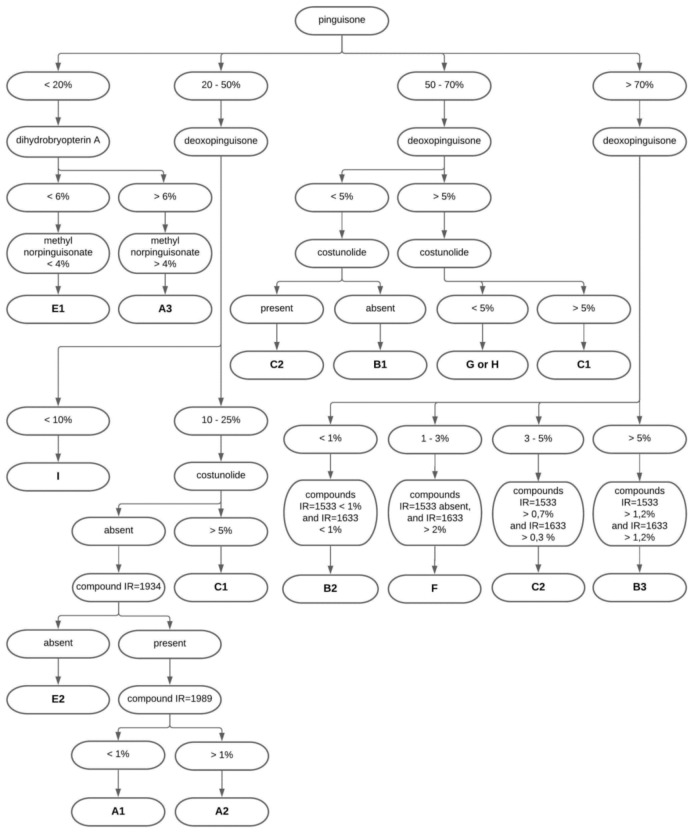
Guide for the identification of cryptic species and genetic lineages of *A. pinguis* based on the content of marker compounds.

**Table 1 molecules-26-01180-t001:** Sampling data of *Aneura* species used for studies and numbers of POZW herbarium vouchers.

Genotype Groups	Sample Code	Collection Place	Geographic Coordinates	Date mm.yyyy	POZW
		***Aneura pinguis*** **cryptic species C**			
C1	C1-1	S Poland, Tatry Mts, NE slope of Skupniów Upłaz Mt.	49°15′ N, 20°00′ E	09.2014	42878
C1	C1-2	S Poland, Tatry Mts, valley of Biały Potok stream	49°16′ N, 19°57′ E	09.2014	42877
C1	C1-3	S Poland, Tatry Mts, Sucha Woda Valley	49°16′ N, 20°01′ E	09.2014	42819a
C1	C1-4	S Poland, Tatry Mts, Sucha Woda Valley	49°16′ N, 20°01′ E	06.2015	42819b
C1	C1-5	SE Poland, Bieszczady Mts, banks of Górna Solinka stream	49°07′ N, 22°29′ E	08.2014	42875
C1	C1-6	NW Poland, Western Pomerania, Słupia river	54°15′ N, 17°29′ E	06.2014	42874
C1	C1-7	NW Poland, Western Pomerania, Lake Garczyn	54°07′ N, 17°54′ E	06.2014	42873
C1	C1-8	NW Poland, Western Pomerania, Lake Kulkówko	54°04′ N, 17°51′ E	06.2014	42912a
C1	C1-9	NW Poland, Western Pomerania, Lake Kulkówko	54°04′ N, 17°51′ E	11.2014	42912b
C1	C1-10	NW Poland, Western Pomerania, Lake Kulkówko	54°04′ N, 17°51′ E	06.2015	42912c
C2	C2-1	SE Poland, Bieszczady Mts, valley of Beskidnik stream	49°06′ N, 22°29′ E	07.2019	42940
C2	C2-2	SE Poland, BieszczadyMts, valley of Górna Solinka stream	49°06′ N, 22°31′ E	07.2014	42792
C2	C2-3	SE Poland, Bieszczady Mts, Ustrzyki Górne, small stream	49°05′ N, 22°38′ E	07.2014	42880
C2	C2-4	SE Poland, Bieszczady Mts, valley of Terebowiec stream	49°06′ N, 22°42′ E	07.2019	42939
		***Aneura pinguis*** **cryptic species E**			
E1	E1-1	S Poland, Tatry Mts, valley of Biały Potok stream	49°16′ N, 19°57′ E	09.2014	42824a
E1	E1-2	S Poland, Tatry Mts, valley of Biały Potok stream	49°16′ N, 19°57′ E	06.2015	42824b
E1	E1-3	S Poland, Tatry Mts, valley of Biały Potok stream	49°16′ N, 19°57′ E	09.2015	42824c
E1	E1-4	S Poland, Tatry Mts, Wielka Sucha Dolina valley	49°16′ N, 19°49′ E	09.2015	40521
E1	E1-5	S Poland, Tatry Mts, Pańszczyca Valley	49°15′ N, 20°02′ E	09.2015	40520
E2	E2-1	S Poland, Pieniny Mts, Limbargowy Potok stream	49°25′ N, 20°21′ E	08.2014	42888a
E2	E2-2	S Poland, Pieniny Mts, Limbargowy Potok stream	49°25′ N, 20°21′ E	08.2015	42888b
E2	E2-3	S Poland, Pieniny Mts, valley Biały Potok stream	49°25′ N, 20°23′ E	09.2019	42938
E2	E2-4	S Poland, Pieniny Mts, Kotłowy Potok stream	49°24′ N, 20°24′ E	08.2018	42937
		***Aneura pinguis*** **cryptic species G**			
G	G-1	SE Poland, Bieszczady Mts, Beskid Pass	49°03′ N, 22°46′ E	07.2014	42896
G	G-2	S Poland, Beskidy Mts, Dejakowy Potok stream	49°36′ N, 19°31′ E	09.2012	42897
G	G-3	NE Poland, Suwałki Lake District, lake Druce	54°06′ N, 23°21′ E	05.2017	42934
G	G-4	NW Poland, Western Pomerania, Lake Małe Płocice	54°05′ N, 17°51′ E	06.2014	42767
		***Aneura pinguis*** **cryptic species H**			
H	H-1	NW Poland, Western Pomerania, Lake Małe Sitno	54°16′ N, 17°30′ E	06. 2018	42936
H	H-2	Central Poland, Wielkopolska, valley of Rurzyca river	53°17′ N, 16°44′ E	10.2014	42898
H	H-3	S Poland, Śląsk, near Katowice	50°14′ N, 19°00′ E	06.2014	42755
H	H-4	NE Poland, Suwałki Lake District, Lake Sejny	54°07′ N, 23°19′ E	05.2019	42935
		***Aneura pinguis*** **cryptic species I**			
I	I-1	NW Poland, Western Pomerania, Lake Małe Oczko	54°03′ N, 18°00′ E	06.2014	42760a
I	I-2	NW Poland, Western Pomerania, Lake Małe Oczko	54°03′ N, 18°00′ E	11.2014	42760b
I	I-3	NW Poland, Western Pomerania, Lake Małe Oczko	54°03′ N, 18°00′ E	06.2015	42760c
I	I-4	NW Poland, Western Pomerania, Lake Duże Sitno	54°16′ N, 17°29′ E	06.2014	42899a
I	I-5	NW Poland, Western Pomerania, Lake Duże Sitno	54°16′ N, 17°29′ E	11.2014	42899b
I	I-6	NW Poland, Western Pomerania, Lake Duże Sitno	54°16′ N, 17°29′ E	06.2015	42899c

**Table 2 molecules-26-01180-t002:** Volatile compounds detected in the analyzed samples of genetic lineage C1 of the *A. pinguis.*

No.	Compounds	RI ^a^	Code ^b^									
			**C1-1**	**C1-2**	**C1-3**	**C1-4**	**C1-5**	**C1-6**	**C1-7**	**C1-8**	**C1-9**	**C1-10**
1	1-Penten-3-ol	673	0.27	0.32	0.49	0.39	0.11	0.09	0.13	0.11	0.19	0.21
2	Pentanal	677	-	-	-	-	-	-	-	-	-	-
3	(*Z*)-2-Penten-1-ol	766	0.09	0.11	0.21	0.22	0.09	0.11	0.10	0.06	0.08	0.12
4	Hexanal	773	2.21	0.69	0.81	1.09	1.97	2.49	1.32	0.84	0.91	0.89
5	(*Z*)-3-Hexen-1-ol	858	1.82	2.41	1.62	1.82	0.89	2.71	1.73	1.43	1.07	0.98
6	1-Hexanol	867	3.39	5.43	2.22	1.73	1.52	1.90	1.48	2.29	2.07	1.87
7	α-Pinene	936	-	-	-	-	-	-	-	-	-	-
8	Camphene	953	-	-	-	-	-	-	-	-	-	-
9	Benzaldehyde	970	-	-	-	-	-	-	-	-	-	-
10	β-Pinene	978	0.39	0.10	0.19	0.21	0.51	0.11	0.39	0.12	0.44	0.23
11	δ-Elemene	1324	-	-	-	-	-	-	-	-	-	-
12	α-Longipinene	1350	0.22	0.02	0.03	0.06	0.12	0.03	0.02	0.08	0.06	0.05
13	α-Ylangene	1373	0.29	0.12	0.22	0.23	0.11	0.12	0.06	0.29	0.13	0.23
14	α-Copaene	1376	-	-	-	-	-	-	-	-	-	-
15	Sativene	1396	0.13	0.20	0.41	0.32	0.29	0.23	0.03	0.07	0.11	0.06
16	β-Caryophyllene	1415	0.69	0.28	0.63	0.43	0.12	0.32	0.18	0.65	0.43	0.52
17	Longifolene	1417	0.89	0.41	1.08	1.06	0.98	0.57	0.49	0.83	0.71	1.20
18	β-Ylangene	1423	0.81	0.73	0.51	0.70	0.54	0.59	0.38	0.62	0.61	0.30
19	Cedrene	1427	2.72	2.22	2.71	3.91	1.20	0.89	0.80	2.39	2.33	2.41
20	204[M]+, 161 (100), 105(85)	1431	-	-	-	-	-	-	-	-	-	-
21	Thujopsene	1435	0.38	0.70	1.31	2.18	2.39	0.06	0.10	0.18	0.10	0.09
22	α-Guaiene	1439	-	-	-	-	-	-	-	-	-	-
23	204[M]+, 93(100), 81(82)	1441	-	-	-	-	-	-	-	-	-	-
24	204[M]+, 93(100), 79(85)	1443	-	-	-	-	-	-	-	-	-	-
25	204[M]+, 119(100), 93(75)	1445	0.38	0.13	0.29	0.52	0.21	0.05	0.06	0.11	0.08	0.72
26	α-Himachalene	1447	0.10	0.05	0.08	0.11	0.04	0.21	0.12	0.22	0.09	0.10
27	204[M]+, 91(100), 105(93)	1453	0.22	0.10	0.08	0.23	0.07	0.21	0.10	0.18	0.08	0.22
28	Alloaromadendrene	1457	0.40	0.04	0.17	0.09	0.32	0.13	0.23	0.07	0.05	0.48
29	γ-Gurjunene	1470	0.12	0.06	0.04	0.39	0.05	0.21	0.09	0.07	0.08	0.41
30	β-Himachalene	1500	0.20	0.04	0.06	0.11	-	-	-	-	-	-
31	204[M]+, 119(100), 91(70)	1503	0.19	0.06	0.07	0.09	0.21	0.09	0.06	0.21	0.06	0.11
32	Cuparene	1505	-	-	-	-	-	-	-	-	-	-
33	δ-Cadinene	1517	-	-	-	-	-	-	-	-	-	-
34	β-Sesquiphellandrene	1520	0.03	0.06	0.07	0.32	0.06	0.07	0.11	0.12	0.06	0.12
35	204[M]+, 119(100), 91(66)	1524	-	-	-	-	-	-	-	-	-	-
36	204[M]+, 69(100), 91(38)	1530	-	-	-	-	-	-	-	-	-	-
37	216[M]+, 109(100), 145(75)	1533	0.99	0.69	1.19	1.09	1.08	4.59	4.78	2.89	2.67	3.11
38	α-Copaene-11-ol	1540	-	-	-	-	-	-	-	-	-	-
39	Longicamphenylone	1559	-	-	-	-	-	-	-	-	-	-
40	Deoxopinguisone	1563	15.69	11.82	14.39	13.49	20.21	11.69	12.42	20.41	17.22	16.70
41	220[M]+, 91(100), 110(90)	1585	-	-	-	-	-	-	-	-	-	-
42	204[M]+, 137(100), 81(60)	1593	-	-	-	-	-	-	-	-	-	-
43	218[M]+, 91(100), 119(57)	1603	0.81	0.62	0.11	0.40	0.13	0.22	0.08	0.59	0.57	0.87
44	220[M]+, 135(100), 91(60)	1620	-	-	-	-	-	-	-	-	-	-
45	220[M]+, 110(100), 91(97)	1626	-	-	-	-	-	-	-	-	-	-
46	220[M]+, 91(100), 79(82)	1633	0.29	2.37	0.69	1.18	0.51	1.38	1.59	0.61	0.82	1.33
47	Longiverbenone	1641	-	-	-	-	-	-	-	-	-	-
48	δ-Cadinol	1646	-	-	-	-	-	-	-	-	-	-
49	Aromadendreneoxide-(2)	1678	-	-	-	-	-	-	-	-	-	-
50	222[M]+, 95(100), 81(75)	1685	0.09	0.22	0.19	0.51	0.63	0.21	0.11	0.41	0.09	0.63
51	220[M]+, 81(100), 67(90)	1687	0.61	0.70	0.78	0.63	0.13	0.32	0.69	0.61	1.02	0.23
52	Pinguisone	1705	56.39	58.12	55.91	55.33	50.28	50.67	51.70	47.08	47.42	51.08
53	Isopinguisone	1712	-	-	-	-	-	-	-	-	-	-
54	Bryopterin A	1735	-	-	-	-	-	-	-	-	-	-
55	Dihydrobryopterin A	1755	0.12	2.99	3.18	1.52	0.12	0.21	0.32	0.42	0.51	0.05
56	Methyl norpinguisonate	1791	-	-	-	-	-	-	-	-	-	-
57	236[M]+, 95(100), 109(85)	1846	-	-	-	-	-	-	-	-	-	-
58	244[M]+, 189(100), 105(15)	1851	-	-	-	-	-	-	-	-	-	-
59	246[M]+, 192(100), 91(43)	1882	-	-	-	-	-	0.08	0.09	0.11	0.12	0.13
60	Costunolide	1914	6.81	6.29	8.72	8.38	12.61	17.99	18.21	14.81	18.09	12.31
61	247[M]+, 147(100), 119(75)	1934	-	-	-	-	-	-	-	-	-	-
62	276[M]+, 108(100), 79(26)	1938	-	-	-	-	-	-	-	-	-	-
63	276[M]+, 108(100), 79(35)	1951	-	-	-	-	-	-	-	-	-	-
64	272[M]+, 81(100), 161(75)	1955	-	-	-	-	-	-	-	-	-	-
65	217[M]+, 108(100), 79(50)	1989	-	-	-	-	-	-	-	-	-	-
66	230[M]+, 108(100), 79(35)	2051	-	-	-	-	-	-	-	-	-	-
	Total		97.74	98.10	98.46	98.74	97.50	98.55	97.97	98.88	98.27	97.76
	% Identified		94.16	93.21	95.06	94.09	94.53	91.40	90.41	93.16	92.76	90.41
	Including:										
	Aliphatics		7.78	8.96	5.35	5.25	4.58	7.30	4.76	4.73	4.32	4.07
	Aromatics		0.00	0.00	0.00	0.00	0.00	0.00	0.00	0.00	0.00	0.00
	Monoterpene hydrocarbons		0.39	0.10	0.19	0.21	0.51	0.11	0.39	0.12	0.44	0.23
	Sesquiterpene hydrocarbons		6.98	4.93	7.32	9.91	6.22	3.43	2.61	5.59	4.76	5.97
	Oxygenated sesquiterpenoids		79.01	79.22	82.20	78.72	83.22	80.56	82.65	82.72	83.24	80.14

- under 0.01%. ^a^ Retention index on Quadex 007-5MS column. ^b^ For abbreviations of samples see Table 1.

**Table 3 molecules-26-01180-t003:** Volatile compounds detected in the analyzed samples of genetic lineage C2 and cryptic species I of the *A. pinguis.*

No.	Compounds	RI ^a^	Code ^b^									
			C2-1	C2-2	C2-3	C2-4	I-1	I-2	I-3	I-4	I-5	I-6
1	1-Penten-3-ol	673	0.09	0.09	0.09	0.08	-	-	-	-	-	-
2	Pentanal	677	-	-	-	-	0.18	0.32	0.21	0.11	0.29	0.12
3	(*Z*)-2-Penten-1-ol	766	0.09	0.10	0.08	0.11	0.11	0.21	0.06	0.12	0.22	0.07
4	Hexanal	773	0.79	0.83	0.63	0.65	1.49	1.12	0.91	1.28	1.64	0.72
5	(*Z*)-3-Hexen-1-ol	858	1.63	1.79	1.22	1.67	0.52	1.01	0.63	0.92	0.71	0.69
6	1-Hexanol	867	3.31	3.23	4.22	3.85	5.81	6.52	5.92	8.41	6.23	4.39
7	α-Pinene	936	-	-	-	-	0.12	0.08	0.11	0.09	0.09	0.10
8	Camphene	953	-	-	-	-	0.07	0.12	0.04	0.12	0.32	0.19
9	Benzaldehyde	970	-	-	-	-	-	-	-	-	-	-
10	β-Pinene	978	0.29	0.37	0.32	0.39	0.21	0.31	0.41	0.20	0.49	0.11
11	δ-Elemene	1324	-	-	-	-	-	-	-	-	-	-
12	α-Longipinene	1350	0.15	0.12	0.13	0.15	0.09	0.10	0.12	0.17	0.13	0.11
13	α-Ylangene	1373	0.11	0.08	0.09	0.10	0.32	0.11	0.31	0.23	0.32	0.29
14	α-Copaene	1376	-	-	-	-	0.10	0.08	0.07	0.11	0.12	0.11
15	Sativene	1396	0.43	0.49	0.23	0.51	0.20	0.22	0.18	0.21	0.22	0.21
16	β-Caryophyllene	1415	0.26	0.23	0.49	0.38	0.04	0.11	0.23	0.11	0.13	0.23
17	Longifolene	1417	0.51	0.47	0.62	0.57	0.31	0.30	0.42	0.21	0.20	0.17
18	β-Ylangene	1423	0.64	0.69	0.59	0.61	0.70	0.51	0.68	0.54	0.61	0.63
19	Cedrene	1427	3.41	3.47	2.21	3.39	0.32	0.41	0.71	0.53	0.49	0.38
20	204[M]+, 161 (100), 105(85)	1431	-	-	-	-	0.10	0.11	0.12	0.13	0.11	0.09
21	Thujopsene	1435	0.09	0.13	0.09	0.12	0.08	0.06	0.09	0.06	0.09	0.11
22	α-Guaiene	1439	-	-	-	-	0.31	0.32	0.40	0.58	0.32	0.31
23	204[M]+, 93(100), 81(82)	1441	-	-	-	-	1.21	0.73	1.07	0.67	0.83	0.92
24	204[M]+, 93(100), 79(85)	1443	-	-	-	-	0.04	0.06	0.05	0.07	0.06	0.05
25	204[M]+, 119(100), 93(75)	1445	0.16	0.12	0.22	0.14	0.06	0.05	0.07	0.06	0.04	0.05
26	α-Himachalene	1447	0.09	0.11	0.12	0.08	0.23	0.11	0.20	0.11	0.08	0.17
27	204[M]+, 91(100), 105(93)	1453	0.32	0.37	0.30	0.41	0.59	0.82	0.44	0.72	0.89	0.59
28	Alloaromadendrene	1457	0.27	0.32	0.09	0.33	2.61	1.73	2.23	1.54	2.53	1.81
29	γ-Gurjunene	1470	0.09	0.12	0.39	0.18	0.10	0.03	0.07	0.09	0.11	0.21
30	β-Himachalene	1500	-	-	-	-	0.12	0.09	0.05	0.06	0.11	0.23
31	204[M]+, 119(100), 91(70)	1503	0.12	0.10	0.06	0.09	0.13	0.23	0.12	0.23	0.11	0.22
32	Cuparene	1505	-	-	-	-	0.23	0.32	0.34	0.27	0.26	0.42
33	δ-Cadinene	1517	-	-	-	-	0.20	0.09	0.22	0.13	0.21	0.17
34	β-Sesquiphellandrene	1520	0.05	0.06	0.04	0.03	0.08	0.04	0.05	0.08	0.06	0.11
35	204[M]+, 119(100), 91(66)	1524	-	-	-	-	-	-	-	-	-	-
36	204[M]+, 69(100), 91(38)	1530	-	-	-	-	4.61	4.89	5.29	4.52	4.71	5.92
37	216[M]+, 109(100), 145(75)	1533	1.18	1.22	1.32	1.27	3.21	2.60	4.38	1.92	2.49	3.32
38	α-Copaene-11-ol	1540	-	-	-	-	0.19	0.43	0.41	0.39	0.20	0.21
39	Longicamphenylone	1559	-	-	-	-	0.12	0.19	0.32	0.11	0.12	0.11
40	Deoxopinguisone	1563	3.59	3.62	4.42	3.98	5.60	5.12	5.54	5.32	5.61	5.68
41	220[M]+, 91(100), 110(90)	1585	-	-	-	-	0.11	0.22	0.23	0.12	0.10	0.09
42	204[M]+, 137(100), 81(60)	1593	-	-	-	-	0.09	0.12	0.11	0.05	0.04	0.08
43	218[M]+, 91(100), 119(57)	1603	6.28	6.33	6.99	6.59	0.06	0.05	0.07	0.11	0.06	0.05
44	220[M]+, 135(100), 91(60)	1620		-	-	-	6.42	4.61	6.69	4.29	4.83	4.52
45	220[M]+, 110(100), 91(97)	1626		-	-	-	0.11	0.10	0.06	0.05	0.04	0.11
46	220[M]+, 91(100), 79(82)	1633	0.10	0.09	0.11	0.12	0.05	0.11	0.21	0.12	0.09	0.10
47	Longiverbenone	1641	-	-	-	-	1.70	1.70	1.60	1.20	1.50	1.90
48	δ-Cadinol	1646	-	-	-	-	-	-	-	-	-	-
49	Aromadendreneoxide-(2)	1678	-	-	-	-	0.11	0.10	0.04	0.12	0.05	0.09
50	222[M]+, 95(100), 81(75)	1685	0.19	0.22	0.20	0.25	2.90	2.43	2.89	3.71	4.32	4.81
51	220[M]+, 81(100), 67(90)	1687	0.49	0.53	0.12	0.42	0.89	0.91	0.60	0.41	0.40	0.89
52	Pinguisone	1705	71.92	71.80	69.29	69.06	43.59	45.22	43.91	44.69	43.71	43.90
53	Isopinguisone	1712	-	-	-	-	0.11	0.05	0.04	0.12	0.20	0.09
54	Bryopterin A	1735	-	-	-	-	0.05	0.09	0.05	0.08	0.11	0.12
55	Dihydrobryopterin A	1755	0.27	0.30	0.23	0.26	0.69	0.32	0.51	0.48	0.32	0.51
56	Methyl norpinguisonate	1791	-	-	-	-	0.20	0.21	0.19	0.39	0.22	0.19
57	236[M]+, 95(100), 109(85)	1846	-	-	-	-	-	-	-	-	-	-
58	244[M]+, 189(100), 105(15)	1851	-	-	-	-	-	-	-	-	-	-
59	246[M]+, 192(100), 91(43)	1882	-	-	-	-	1.90	2.11	1.99	2.08	2.41	2.12
60	Costunolide	1914	1.44	1.49	2.89	1.51	3.82	5.91	1.80	4.30	3.59	3.85
61	247[M]+, 147(100), 119(75)	1934	-	-	-	-	0.33	0.42	0.69	0.44	0.33	0.42
62	276[M]+, 108(100), 79(26)	1938	-	-	-	-	-	-	-	-	-	-
63	276[M]+, 108(100), 79(35)	1951	-	-	-	-	-	-	-	-	-	-
64	272[M]+, 81(100), 161(75)	1955	-	-	-	-	1.39	1.41	1.62	1.74	1.54	1.70
65	217[M]+, 108(100), 79(50)	1989	-	-	-	-	-	-	-	-	-	-
66	230[M]+, 108(100), 79(35)	2051	-	-	-	-	2.40	2.52	2.61	2.59	2.71	2.33
	Total		98.36	98.89	97.80	97.30	97.33	98.16	98.38	97.51	97.72	97.09
	% Identified		89.52	89.91	88.48	88.01	70.73	73.66	69.07	73.48	71.61	68.71
	Including:									
	Aliphatics		5.91	6.04	6.24	6.36	8.11	9.18	7.73	10.84	9.09	5.99
	Aromatics		0.00	0.00	0.00	0.00	0.00	0.00	0.00	0.00	0.00	0.00
	Monoterpene hydrocarbons		0.29	0.37	0.32	0.39	0.40	0.51	0.56	0.41	0.90	0.40
	Sesquiterpene hydrocarbons		6.10	6.29	5.09	6.45	6.04	4.63	6.37	5.03	5.99	5.67
	Oxygenated sesquiterpenoids		77.22	77.21	76.83	74.81	56.18	59.34	54.41	57.20	55.63	56.65

- under 0.01%. ^a^ Retention index on Quadex 007-5MS column. ^b^ For abbreviations of samples see Table 1.

**Table 4 molecules-26-01180-t004:** Volatile compounds detected in the analyzed samples of cryptic species E (genetic lineage E1 and E2) of the *A. pinguis.*

No.	Compounds	RI ^a^	Code ^b^								
			E1-1	E1-2	E1-3	E1-4	E1-5	E2-1	E2-2	E2-3	E2-4
1	1-Penten-3-ol	673	0.11	0.12	0.23	0.12	0.04	0.99	0.61	0.70	0.87
2	Pentanal	677	-	-	-	-	-	-	-	-	-
3	(*Z*)-2-Penten-1-ol	766	0.06	0.12	0.06	0.12	0.06	0.41	0.39	0.45	0.37
4	Hexanal	773	0.12	0.13	0.09	0.21	0.32	1.02	0.92	0.97	1.09
5	(*Z*)-3-Hexen-1-ol	858	0.41	0.52	0.07	0.32	0.41	1.59	1.41	1.45	1.53
6	1-Hexanol	867	0.92	0.91	0.62	0.69	1.33	1.99	1.91	2.02	1.95
7	α-Pinene	936	0.33	0.06	0.52	0.13	0.11	0.63	0.79	0.59	0.83
8	Camphene	953	0.06	0.41	0.11	0.10	0.12	0.09	0.10	0.08	0.09
9	Benzaldehyde	970	-	-	-	-	-	-	-	-	-
10	β-Pinene	978	0.09	0.21	0.07	0.12	0.11	0.32	0.22	0.37	0.25
11	δ-Elemene	1324	-	-	-	-	-	-	-	-	-
12	α-Longipinene	1350	0.22	0.05	0.04	0.06	0.11	0.06	0.06	0.04	0.09
13	α-Ylangene	1373	0.11	0.12	0.32	0.08	0.19	0.06	0.07	0.07	0.08
14	α-Copaene	1376	0.10	0.05	0.07	0.04	0.21	0.11	0.09	0.07	0.13
15	Sativene	1396	0.92	1.28	1.41	1.21	1.07	1.32	0.79	1.21	1.26
16	β-Caryophyllene	1415	0.22	0.11	0.09	0.10	0.19	0.17	0.42	0.26	0.37
17	Longifolene	1417	3.51	3.79	4.12	4.22	4.31	1.02	0.83	0.92	0.86
18	β-Ylangene	1423	5.30	3.08	4.48	3.71	4.82	0.50	0.48	0.43	0.53
19	Cedrene	1427	1.72	1.52	1.48	1.31	1.42	0.11	0.11	0.09	0.07
20	204[M]+, 161 (100), 105(85)	1431	0.41	0.21	0.28	0.17	0.52	0.73	0.51	0.64	0.57
21	Thujopsene	1435	1.83	1.60	2.81	2.27	2.53	0.88	1.08	1.14	0.97
22	α-Guaiene	1439	0.49	1.17	0.77	0.62	0.42	1.09	0.89	0.92	0.94
23	204[M]+, 93(100), 81(82)	1441	7.19	7.69	6.89	7.88	8.43	1.63	1.84	1.73	1.91
24	204[M]+, 93(100), 79(85)	1443	0.49	0.08	0.38	0.08	0.09	0.22	0.12	0.26	0.18
25	204[M]+, 119(100), 93(75)	1445	1.72	0.81	1.23	1.72	1.50	0.08	0.09	0.07	0.06
26	α-Himachalene	1447	1.98	2.81	1.82	2.51	2.83	0.19	0.42	0.23	0.31
27	204[M]+, 91(100), 105(93)	1453	-	-	-	-	-	0.51	0.92	0.85	0.62
28	Alloaromadendrene	1457	10.51	11.22	11.69	10.72	10.22	0.13	0.31	0.18	0.34
29	γ-Gurjunene	1470	-	-	-	-	-	0.19	0.18	0.15	0.23
30	β-Himachalene	1500	5.69	6.72	4.91	6.42	6.43	0.09	0.09	0.08	0.07
31	204[M]+, 119(100), 91(70)	1503	0.69	0.51	0.80	1.08	0.99	0.08	0.09	0.09	0.11
32	Cuparene	1505	1.20	0.91	1.31	1.90	0.71	0.10	0.11	0.07	0.13
33	δ-Cadinene	1517	0.29	0.22	0.51	0.18	0.32	0.31	0.30	0.34	0.27
34	β-Sesquiphellandrene	1520	0.20	0.11	0.30	0.11	0.33	0.10	0.11	0.07	0.09
35	204[M]+, 119(100), 91(66)	1524	0.18	0.20	0.22	0.19	0.29	-	-	-	-
36	204[M]+, 69(100), 91(38)	1530	0.72	0.80	0.48	0.41	0.83	0.11	0.10	0.13	0.09
37	216[M]+, 109(100), 145(75)	1533	0.19	0.23	0.29	0.30	0.32	0.33	0.53	0.58	0.41
38	α-Copaene-11-ol	1540	0.04	0.08	0.07	0.41	0.11	0.06	0.05	0.03	0.05
39	Longicamphenylone	1559	1.22	2.31	1.47	1.02	0.79	0.20	0.30	0.24	0.29
40	Deoxopinguisone	1563	9.11	7.92	6.52	7.00	6.72	22.61	24.60	23.86	22.37
41	220[M]+, 91(100), 110(90)	1585	0.09	0.60	0.42	0.73	0.44	-	-	-	-
42	204[M]+, 137(100), 81(60)	1593	1.11	0.87	1.20	0.93	1.11	0.09	0.08	0.09	0.07
43	218[M]+, 91(100), 119(57)	1603	1.53	1.42	1.28	1.83	1.33	-	-	-	-
44	220[M]+, 135(100), 91(60)	1620	5.10	5.37	5.23	6.31	5.88	0.31	0.19	0.34	0.28
45	220[M]+, 110(100), 91(97)	1626	0.11	0.09	0.10	0.23	0.29	0.11	0.40	0.36	0.19
46	220[M]+, 91(100), 79(82)	1633	2.29	1.80	1.73	2.09	2.20	0.07	0.19	0.11	0.21
47	Longiverbenone	1641	1.19	1.33	1.90	1.93	1.62	0.52	0.88	0.79	0.64
48	δ-Cadinol	1646	-	-	-	-	-	-	-	-	-
49	Aromadendreneoxide-(2)	1678	2.60	2.87	2.81	2.33	2.41	0.09	0.20	0.15	0.19
50	222[M]+, 95(100), 81(75)	1685	2.56	2.07	2.69	2.93	2.72	0.41	0.31	0.27	0.39
51	220[M]+, 81(100), 67(90)	1687	1.42	0.71	1.98	0.81	0.58	0.11	0.21	0.17	0.09
52	Pinguisone	1705	8.44	11.95	9.08	7.89	7.07	35.42	31.08	32.59	33.63
53	Isopinguisone	1712	4.52	3.52	3.41	3.20	3.73	6.42	5.60	5.82	6.14
54	Bryopterin A	1735	0.12	0.19	0.06	0.10	0.41	4.08	4.28	3.89	4.18
55	Dihydrobryopterin A	1755	4.32	4.51	4.92	5.42	4.29	9.89	10.01	10.85	10.19
56	Methyl norpinguisonate	1791	2.41	1.82	1.93	1.72	1.70	0.19	0.08	0.12	0.07
57	236[M]+, 95(100), 109(85)	1846	-	-	-	-	-	-	-	-	-
58	244[M]+, 189(100), 105(15)	1851	-	-	-	-	-	-	-	-	-
59	246[M]+, 192(100), 91(43)	1882	-	-	-	-	-	-	-	-	-
60	Costunolide	1914	-	-	-	-	-	-	-	-	-
61	247[M]+, 147(100), 119(75)	1934	-	-	-	-	-	-	-	-	-
62	276[M]+, 108(100), 79(26)	1938	0.05	0.11	0.12	0.05	0.09	0.11	0.10	0.09	0.08
63	276[M]+, 108(100), 79(35)	1951	0.91	0.79	1.42	1.43	1.62	0.40	0.41	0.29	0.34
64	272[M]+, 81(100), 161(75)	1955	0.04	0.03	0.05	0.21	0.08	0.11	0.11	0.12	0.09
65	217[M]+, 108(100), 79(50)	1989	0.40	0.52	0.37	0.38	0.40	0.42	0.71	0.34	0.64
66	230[M]+, 108(100), 79(35)	2051	0.18	0.19	0.42	0.21	0.43	0.11	0.49	0.21	0.13
	Total		97.74	98.84	97.65	98.26	97.60	98.89	97.17	97.98	97.93
	% Identified		70.36	73.74	70.07	68.29	67.46	92.95	89.77	91.24	91.47
	Including:									
	Aliphatics		1.62	1.80	1.07	1.46	2.16	6.00	5.24	5.59	5.81
	Aromatics		0.00	0.00	0.00	0.00	0.00	0.00	0.00	0.00	0.00
	Monoterpene hydrocarbons		0.48	0.68	0.70	0.35	0.34	1.04	1.11	1.04	1.17
	Sesquiterpene hydrocarbons		34.29	34.76	36.13	35.46	36.11	6.43	6.34	6.27	6.74
	Oxygenated sesquiterpenoids		33.97	36.50	32.17	31.02	28.85	79.48	77.08	78.34	77.75

- under 0.01%. ^a^ Retention index on Quadex 007-5MS column. ^b^ For abbreviations of samples see Table 1.

**Table 5 molecules-26-01180-t005:** Volatile compounds detected in the analyzed samples of cryptic species G and H of the *A. pinguis.*

No.	Compounds	RI ^a^	Code ^b^							
			G-1	G-2	G-3	G-4	H-1	H-2	H-3	H-4
1	1-Penten-3-ol	673	0.04	0.05	0.06	0.06	0.05	0.05	0.06	0.04
2	Pentanal	677	0.11	0.08	0.10	0.09	0.08	0.08	0.09	0.11
3	(*Z*)-2-Penten-1-ol	766	-	-	-	-	-	-	-	-
4	Hexanal	773	0.23	0.15	0.17	0.12	0.23	0.19	0.21	0.25
5	(*Z*)-3-Hexen-1-ol	858	0.30	0.07	0.11	0.09	0.13	0.11	0.22	0.09
6	1-Hexanol	867	0.89	1.01	1.07	1.19	1.55	1.51	1.42	1.48
7	α-Pinene	936	0.21	0.15	0.14	0.11	0.10	0.12	0.08	0.09
8	Camphene	953	0.12	0.06	0.09	0.07	0.06	0.06	0.05	0.03
9	Benzaldehyde	970	-	-	-	-	-	-	-	-
10	β-Pinene	978	0.79	0.72	0.59	0.61	0.41	0.44	0.52	0.48
11	δ-Elemene	1324	-	-	-	-	-	-	-	-
12	α-Longipinene	1350	0.31	0.47	0.52	0.59	0.63	0.60	0.41	0.56
13	α-Ylangene	1373	0.11	0.07	0.04	0.05	0.15	0.13	0.09	0.08
14	α-Copaene	1376	0.51	0.47	0.39	0.41	0.44	0.42	0.43	0.47
15	Sativene	1396	0.51	0.59	0.65	0.62	0.76	0.73	0.64	0.68
16	β-Caryophyllene	1415	0.34	0.37	0.45	0.42	0.18	0.24	0.36	0.27
17	Longifolene	1417	1.27	1.25	1.19	1.23	1.21	1.32	0.99	1.19
18	β-Ylangene	1423	1.21	1.44	1.55	1.53	1.57	1.62	1.11	1.23
19	Cedrene	1427	0.11	0.14	0.10	0.12	0.08	0.08	0.09	0.07
20	204[M]+, 161 (100), 105(85)	1431	0.72	0.83	0.72	0.81	0.91	0.93	1.01	0.96
21	Thujopsene	1435	1.69	1.82	1.73	1.93	2.32	2.37	1.69	2.09
22	α-Guaiene	1439	0.21	0.22	0.16	0.18	0.18	0.22	0.09	0.15
23	204[M]+, 93(100), 81(82)	1441	0.23	0.19	0.16	0.14	0.25	0.23	0.12	0.19
24	204[M]+, 93(100), 79(85)	1443	0.21	0.14	0.11	0.12	0.20	0.22	0.13	0.21
25	204[M]+, 119(100), 93(75)	1445	0.11	0.08	0.11	0.09	0.15	0.12	0.13	0.14
26	α-Himachalene	1447	0.12	0.24	0.27	0.33	0.28	0.31	0.22	0.24
27	204[M]+, 91(100), 105(93)	1453	1.07	1.44	1.52	1.65	1.61	1.58	1.83	1.78
28	Alloaromadendrene	1457	1.40	1.23	1.18	1.11	1.14	1.12	1.32	1.08
29	γ-Gurjunene	1470	0.48	0.51	0.39	0.42	0.41	0.38	0.39	0.45
30	β-Himachalene	1500	0.21	0.22	0.17	0.19	0.19	0.21	0.09	0.13
31	204[M]+, 119(100), 91(70)	1503	1.19	1.01	0.93	0.99	0.98	1.10	1.04	0.14
32	Cuparene	1505	1.37	1.41	1.32	1.29	1.97	1.93	1.42	1.57
33	δ-Cadinene	1517	0.11	0.18	0.16	0.21	0.18	0.10	0.70	0.49
34	β-Sesquiphellandrene	1520	0.09	0.15	0.19	0.20	0.05	0.05	0.07	0.03
35	204[M]+, 119(100), 91(66)	1524	-	-	-	-	-	-	-	-
36	204[M]+, 69(100), 91(38)	1530	0.12	0.11	0.08	0.09	0.08	0.11	0.06	0.12
37	216[M]+, 109(100), 145(75)	1533	2.62	2.43	2.51	2.32	1.69	1.79	1.38	1.63
38	α-Copaene-11-ol	1540	0.22	0.25	0.31	0.29	0.13	0.11	0.12	0.10
39	Longicamphenylone	1559	0.29	0.33	0.37	0.42	0.17	0.22	0.32	0.26
40	Deoxopinguisone	1563	8.99	8.82	8.92	8.71	8.27	8.22	8.31	8.08
41	220[M]+, 91(100), 110(90)	1585	0.11	0.09	0.15	0.12	0.16	0.21	0.09	0.08
42	204[M]+, 137(100), 81(60)	1593	-	-	-	-	-	-	-	-
43	218[M]+, 91(100), 119(57)	1603	0.32	0.24	0.18	0.21	0.36	0.42	0.22	0.26
44	220[M]+, 135(100), 91(60)	1620	0.49	0.47	0.39	0.42	0.91	0.80	0.42	1.01
45	220[M]+, 110(100), 91(97)	1626	0.22	0.12	0.11	0.09	0.13	0.10	0.09	0.07
46	220[M]+, 91(100), 79(82)	1633	0.08	0.07	0.14	0.11	0.08	0.12	0.09	0.07
47	Longiverbenone	1641	-	-	-	-	-	-	-	-
48	δ-Cadinol	1646	1.72	1.77	1.86	1.82	1.71	1.69	1.52	1.58
49	Aromadendreneoxide-(2)	1678	-	-	-	-	-	-	-	-
50	222[M]+, 95(100), 81(75)	1685	0.32	0.24	0.27	0.22	0.37	0.41	0.32	0.43
51	220[M]+, 81(100), 67(90)	1687	3.11	1.83	1.78	1.72	2.01	2.41	1.79	1.93
52	Pinguisone	1705	52.12	53.98	54.21	54.38	56.41	55.50	57.29	56.43
53	Isopinguisone	1712	-			-		-	-	-
54	Bryopterin A	1735	0.19	0.18	0.22	0.20	0.07	0.09	0.10	0.14
55	Dihydrobryopterin A	1755	3.10	4.01	4.74	4.82	3.51	3.61	3.69	3.44
56	Methyl norpinguisonate	1791	0.80	1.32	1.27	1.29	1.20	1.14	1.43	1.09
57	236[M]+, 95(100), 109(85)	1846	-	-	-	-	-	-	-	-
58	244[M]+, 189(100), 105(15)	1851	-	-	-	-	-	-	-	-
59	246[M]+, 192(100), 91(43)	1882	0.09	0.07	0.09	0.10	0.08	0.11	0.12	0.15
60	Costunolide	1914	2.03	2.89	2.97	3.08	2.12	2.06	3.50	2.73
61	247[M]+, 147(100), 119(75)	1934	0.12	0.17	0.19	0.17	0.25	0.22	0.21	0.31
62	276[M]+, 108(100), 79(26)	1938	-	-	-	-	-	-	-	-
63	276[M]+, 108(100), 79(35)	1951	-	-	-	-	-	-	-	-
64	272[M]+, 81(100), 161(75)	1955	-	-	-	-	-	-	-	-
65	217[M]+, 108(100), 79(50)	1989	-	-	-	-	-	-	-	-
66	230[M]+, 108(100), 79(35)	2051	0.04	0.05	0.07	0.06	0.24	0.21	0.14	0.16
	Total		93.37	96.20	97.17	97.61	98.40	98.12	98.23	96.84
	% Identified		82.20	86.62	87.66	88.18	87.94	87.03	89.04	87.20
	Including:								
	Aliphatics		1.57	1.51	1.36	1.55	2.04	1.94	2.00	1.97
	Aromatics		0.00	0.00	0.00	0.00	0.00	0.00	0.00	0.00
	Monoterpene hydrocarbons		1.12	0.93	0.82	0.79	0.57	0.62	0.65	0.60
	Sesquiterpene hydrocarbons		10.05	10.78	10.46	10.83	11.74	11.83	10.11	10.78
	Oxygenated sesquiterpenoids		69.46	73.55	74.87	75.01	73.59	72.64	76.28	73.85

- under 0.01%. ^a^ Retention index on Quadex 007-5MS column. ^b^ For abbreviations of samples see Table 1.

## Data Availability

The data presented in this study is available within the article.

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
