# Peer review of "Chemical Fingerprinting of Cryptic Species and Genetic Lineages of Aneura pinguis (L.) Dumort. (Marchantiophyta, Metzgeriidae)"

_molecules, 2021, doi:10.3390/molecules26041180_

Round 1

Reviewer 1 Report

  1. The article carries out a study of the volatile chemical components of cryptic species (species that are very similar in appearance, but for reproductive purposes they are isolated from each other, and it is used as a tool to understand speciation processes in evolutionary biology ) of the species A. pinguis and whose objective is to develop an efficient method for the characterization of said species using these compounds as chemical markers. Likewise, a guide is prepared to indicate how these chemical markers can be used to carry out a rapid identification of the genetic lineages of said species and thus be able to establish phylogenetic relationships between the cryptic species and the genetic lineages of A. pinguis.

  1. The main interest of this work lies in the exhaustive study of the volatile components of the different cryptic species based on previous studies that the composition of volatile compounds is determined primarily by genetics. The study of this composition allows them to determine nine compounds that they use as markers. And based on these data, create a guide. It would be advisable to include the structures of the compounds used as chemical markers.

  1. A brief introduction on the specific problems of the cryptic species of A. pinguis would be interesting to justify the importance of the chemistry of the compounds as a support tool for taxonomy.

  1. Correct in paragraph 223 costunlide for costunolide.

Author Response

The following changes were made to the manuscript:

Reviewer 1 – the changes made are in blue

- in introduction a brief explanation of A. pinguis cryptic species has been added (point 3)

- costunolide was corrected (point 4)

Reviewer 2 Report

This manuscript describes the main chemical markers of Aneura pinguis and their genetic lineage. I recommend authors to revised manusucript keeping following comments in mind.

  1. Complete scientific name of species "Aneura pinguis (L.) Dumort." should be provided in title, abstract.
  2. Line 74, A. maxima please make in italics and provide authority name for plant scientific names when mentioned for the first time for all species.
  3. Line 113, A. Pinguis: correct to A. pinguis
  4. Table 1, does POZW refer to GenBank accession numbers?
  5. Figure 4 is too small and difficult to see.

Author Response

The following changes were made to the manuscript:

Reviewer 2 – the changes made are in green

– in the legend of Table 1 phrase “and GenBank accession numbers” was deleted and an explanation of the sequence availability is provided at the end of paragraph 3.1. (point 4)

- the font size in Figure 4 has been increased to improve its readability (point 5)

Reviewer 3 Report

I commend the authors for their interesting paper, however, I need some modifications their:

The title: What you did here is called "chemical fingerprint" or chemical identification, so the title need to point this interesting technique to be 

"Chemical fingerprinting of cryptic species and genetic lineages of Aneura pinguis (Marchantiophyta, Metzgeriidae)"

Introduction: you may minimize the introduction, I feel its long. 

results: need to be titled as results and discussion. 

Discussion: I feel some parts need additional discussion. 

The conclusion is ok 

Author Response

The following changes were made to the manuscript:

Reviewer 3 – the changes made are in red

- we changed the title as proposed by the Reviewer, the new title is better.

- at the reviewer's request, we have shortened the introduction

Round 2

Reviewer 2 Report

Authors have made necessary revision.

Reviewer 3 Report

Accepted for me.